# Calcium isotope evidence for early Archaean carbonates and subduction of oceanic crust

Michael A. Antonelli [1,5✉], Jillian Kendrick[2], Chris Yakymchuk[2], Martin Guitreau [3], Tushar Mittal[4] & Frédéric Moynier[1]

Continents are unique to Earth and played a role in coevolution of the atmosphere, hydrosphere, and biosphere. Debate exists, however, regarding continent formation and the onset of subduction-driven plate tectonics. We present Ca isotope and trace-element data from modern and ancient (4.0 to 2.8 Ga) granitoids and phase equilibrium models indicating that Ca isotope fractionations are dominantly controlled by geothermal gradients. The results require gradients of 500–750 °C/GPa, as found in modern (hot) subduction-zones and consistent with the operation of subduction throughout the Archaean. Two granitoids from the Nuvvuagittuq Supracrustal Belt, Canada, however, cannot be explained through magmatic processes. Their isotopic signatures were likely inherited from carbonate sediments. These samples (> 3.8 Ga) predate the oldest known carbonates preserved in the rock record and confirm that carbonate precipitation in Eoarchaean oceans provided an important sink for atmospheric $CO_2$. Our results suggest that subduction-driven plate tectonic processes started prior to ~3.8 Ga.

---

[1] Institut de Physiques du Globe de Paris, Université de Paris, CNRS, UMR 7154, Paris, France. [2] Department of Earth and Environmental Sciences, University of Waterloo, Waterloo, ON, Canada. [3] Université Clermont Auvergne, CNRS, IRD, OPGC, Laboratoire Magma et Volcans, F- 63000 Clermont Ferrand, France. [4] Department of Earth, Planetary and Atmospheric Sciences, Massachusetts Institute of Technology, Cambridge, MA, USA. [5] Present address: Department of Earth Sciences, Institute of Geochemistry and Petrology, ETH Zurich, Zurich, Switzerland. ✉email: mantonelli@berkeley.edu

The mechanisms and timing of continental crust formation are important to understand because they are intimately tied to (i) the history of plate tectonics, (ii) the chemical evolution of the atmosphere and oceans, and (iii) the proliferation of life on Earth[1–3]. There is still much controversy, however, surrounding the formation of continents and the timing for onset of subduction-driven plate tectonics, which is partly due to the difficulty of interpreting Earth's scarce early rock record. Archean continental crust (>2.5 Ga) accounts for only ~5% of Earth's modern surface and is dominated by "granite-greenstone belts" that consist of subordinate metabasalts (~20%) and silicic plutonic rocks known as tonalite–trondhjemite–granodiorite (TTG) suites[4]. TTG suites no longer form in the present day as a consequence of lower mantle temperatures, yet adakites, which are formed during rare melting of subducting hot/young oceanic crust[5], are the closest modern analogs for Archean TTGs[4,6].

TTG suites unequivocally represent Earth's earliest preserved continental crust, yet whether or not subduction of oceanic crust is required for their formation is still debated. Other hypotheses suggest that TTGs derive from processes such as melting of hydrated mafic rocks at the base of thick oceanic plateaus[4,7] or extensive fractional crystallization of basaltic magmas in the mid-to-lower crust[8]. Using Si isotopes, recent studies suggest that surface-derived materials were recycled into the sources of TTGs[9,10], thus favoring a horizontal tectonic scenario[11]. Yet, trace-element ratios, which are typically used to infer $PT$ conditions for TTG petrogenesis, can lead to ambiguous results due to (i) the competing effects of temperature and pressure, (ii) the potential for open-system behavior[12], and (iii) the similar trace-element signatures for restitic garnet (high-pressure) and hornblende (low-pressure), leading to significant debates regarding the depths of TTG generation[13]. Thus, it can still be questioned whether surficial materials (e.g., chert) were incorporated into TTG source rocks through subduction or through other processes such as burial during top–down construction of oceanic plateaus[7].

Here, we develop a stable Ca isotope proxy that can constrain the apparent geothermal gradients ($dT/dP$) along which TTG magmas were generated and show that these can be used to discriminate between different geodynamic settings for the formation of ancient continental crust. Calcium isotopes are ideal for investigating TTG petrogenesis because they (i) are sensitive to magmatic processes and source variations, (ii) commonly equilibrate in plutonic settings, (iii) have well-defined equilibrium fractionation factors, and (iv) are insensitive to redox effects[14].

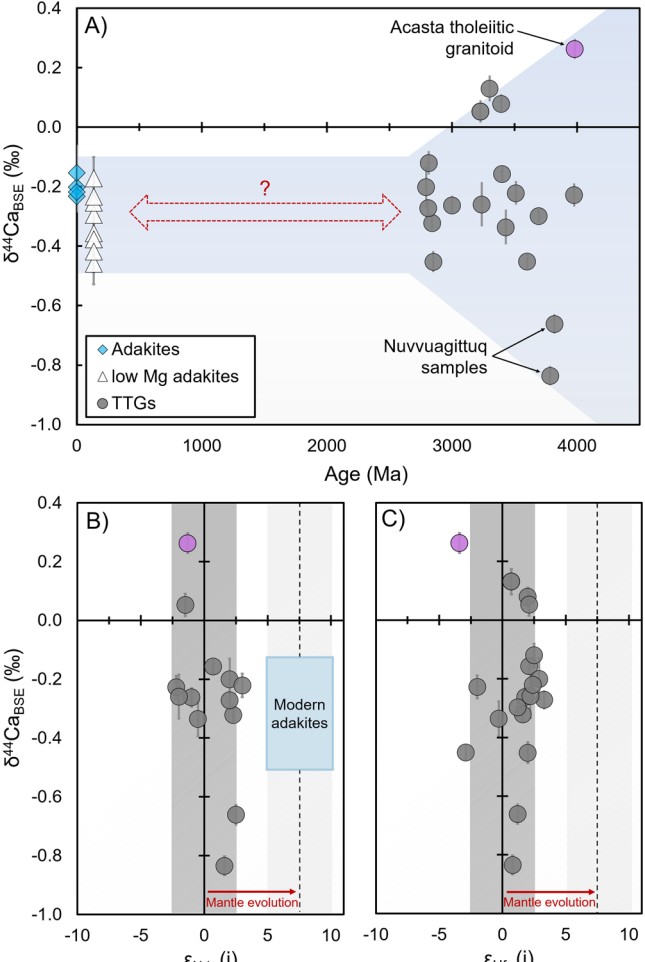

**Fig. 1 Ca isotope data through time and initial radiogenic isotope values[11] for our samples.** A $\delta^{44}$Ca vs. age for TTG/adakite samples and Acasta tholeiitic granitoid (pink circle) analyzed in this study and low Mg adakites[15] ($n = 32$). **B** $\delta^{44}$Ca vs. initial $\varepsilon_{Nd}$ ($n = 13$) for Archean samples. **C** $\delta^{44}$Ca vs. initial $\varepsilon_{Hf}$ ($n = 19$) for Archean samples. The mantle-like isotopic signatures suggest that assimilation of pre-existing crust was not an important process in our samples. Error bars represent 2SE uncertainties on our Ca isotope measurements. TTG tonalite–trondhjemite–granodiorite suite.

## Results and discussion

**Ca isotope and trace-element data**. We report $\delta^{44}$Ca values [defined as $(^{44}Ca/^{40}Ca)_{sample}/(^{44}Ca/^{40}Ca)_{BSE} - 1$, and reported in per mil (‰) relative to bulk-silicate Earth (BSE)] and trace-element abundances [see "Methods," Supplementary Table 1] in modern adakites from the Austral Volcanic Zone ($n = 4$), an Eoarchean tholeiitic granitoid ($n = 1$), and well-characterized TTG samples from around the globe ($n = 18$) ranging in age from 4.0 to 2.8 Ga (Supplementary Note 1 and Supplementary Data 2). The Archean samples have been previously analyzed for Si, Hf, and Nd isotopes and were carefully selected for minimal alteration from a much larger sample set[9,11]. We find relatively large variations in $\delta^{44}$Ca values (>1‰, Supplementary Data 1), with TTG samples ranging from −0.9‰ to +0.2‰ and modern adakite samples clustering tightly at −0.2‰ (Fig. 1a). Most TTG samples fall between −0.2‰ and −0.5‰ ($n = 13$), similar to Mesozoic adakites[15], with a smaller group (including the tholeiitic granitoid, $n = 4$) displaying $\delta^{44}$Ca higher than BSE (>0‰) and samples from the Nuvvuagittuq Supracrustal Belt, Canada (NSB, $n = 2$) with $\delta^{44}$Ca lower than −0.7‰ (Supplementary Data 3).

Initial neodymium and hafnium isotopic compositions have limited ranges consistent with juvenile (mantle) sources and do not correlate with $\delta^{44}$Ca (Fig. 1b, c), suggesting that assimilation of older crust did not influence isotopic signatures. We find that a majority of TTGs and adakites have $\delta^{44}$Ca values that are anticorrelated with trace-element ratios associated with increasing residual garnet in source rocks at higher pressures (Supplementary Fig. 1). Since garnet is predicted to have the highest $\delta^{44}$Ca among Ca-rich minerals[14], this observation suggests that Ca isotopes are sensitive to both temperature and to pressure-dependent changes in residual mineralogy.

**Phase-equilibrium modeling**. In order to best address the opposite isotopic fractionation effects of increasing pressures and temperatures (Supplementary Fig. 2a), and to account for the complex changes in mineral proportions and compositions in TTG source rocks during progressive melting (Supplementary Fig. 2b, c), we have incorporated equilibrium Ca isotope fractionation into phase-equilibrium models for TTG petrogenesis[12] (see "Methods," Supplementary Notes 2 and 3). In Fig. 2a, we

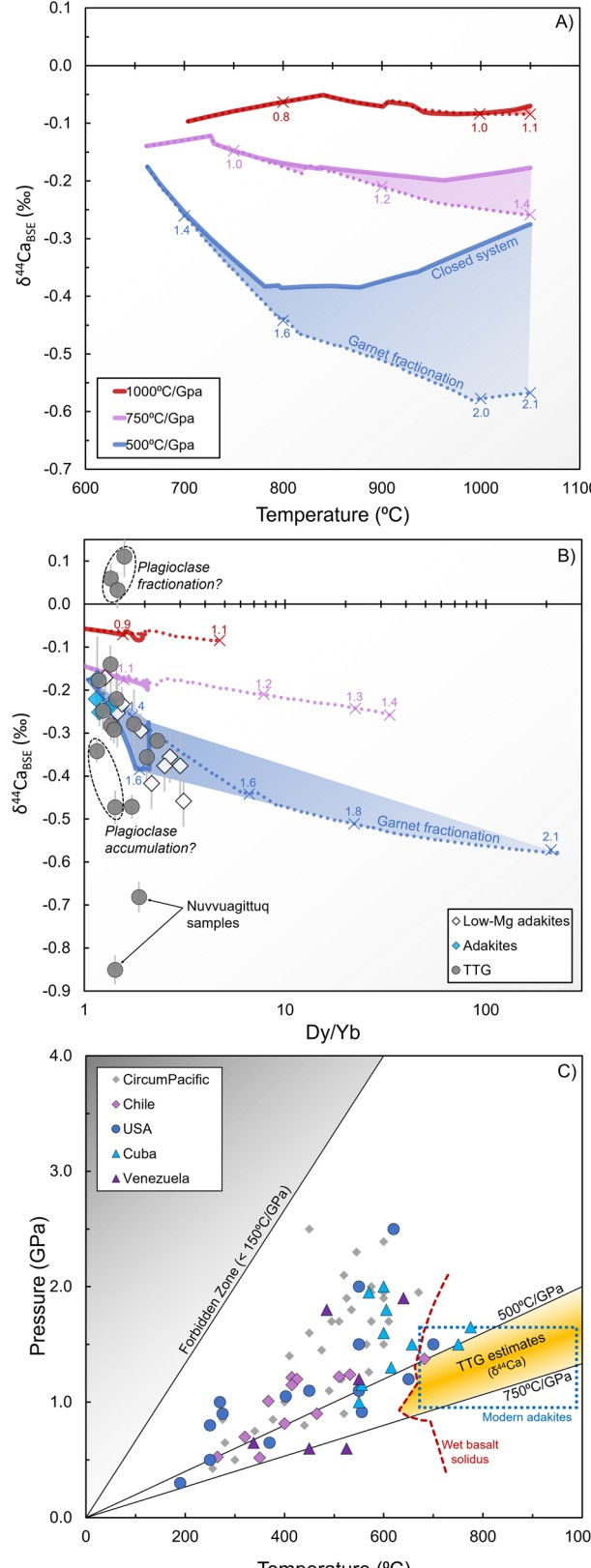

**Fig. 2 Phase-equilibrium model results for depleted Archean tholeiite (DAT) and comparisons with measured δ⁴⁴Ca and modern subduction-zone *PT* estimates. A** Effect of geothermal gradient on δ⁴⁴Ca. **B** Model results and data (n = 32) for δ⁴⁴Ca vs. Dy/Yb (a proxy for residual garnet); colored numbers correspond to pressure (in GPa). See Supplementary Note 5 and Supplementary Fig. 5 for discussion of plagioclase fractionation and accumulation effects. Error bars represent 2SE uncertainties on our Ca isotope measurements. **C** *PT* estimates for modern hot subduction-zone assemblages [eclogite and blueschist data[16] (n = 76)], modern adakites [dotted blue rectangle[6]], and Archean TTG samples (orange field, this study). TTG tonalite–trondhjemite–granodiorite suite.

isotopic fractionation due to increasing residual garnet), with the largest effects predicted along the lowest d*T*/d*P* path (500 °C/GPa). We focus our discussion on models that use a depleted Archean tholeiite (DAT) protolith because they provide the best match to our measured major and trace-element data (Supplementary Fig. 3)[12]. Broadly similar results, however, are obtained for enriched Archean tholeiite (Supplementary Fig. 4), suggesting that our models are relatively insensitive to variations in protolith composition.

**Geothermal gradient constraints in Archean granitoids.** In Fig. 2b, we compare data and model results for δ⁴⁴Ca vs. Dy/Yb (a trace-element proxy for restitic garnet). In concert with δ⁴⁴Ca data and model results for other common trace-element proxies, we find that most TTGs, along with modern and Mesozoic adakites, are best explained by d*T*/d*P* paths between 500 and 750 °C/GPa. The required geothermal gradients are similar to those recorded in hot "modern" (<750 Ma) subduction-zone assemblages[16] but with overall higher temperatures that enable direct melting of mafic (oceanic) crust. Given that our data best agree with closed-system model results, this suggests that TTG melts may have initially encountered thermal or rheological impediments to melt loss[17]. Using the wet-basalt (DAT) solidus[12] as a lower temperature limit (Fig. 2c), our results agree with previous predictions for TTG petrogenesis (800–1000 °C at 1–2 GPa)[18]. These results are also similar to *PT* estimates for modern adakites[6], which are typically formed through subduction of hot/young oceanic crust[5] (Supplementary Note 4b). Melting at the base of thickened oceanic plateaus, on the other hand, is predicted to occur at higher d*T*/d*P* [>700 °C/GPa[18,19], Supplementary Note 4a]. The Ca isotope constraints [along with recent geophysical models[20]] therefore suggest that Archean TTGs most likely formed through (hot) subduction.

Although the results point toward melting of hydrated oceanic crust as the source of TTG magmas, slightly negative europium anomalies in a number of samples point toward a role for plagioclase fractionation within TTG plutons after emplacement[4,8]. Given that plagioclase is the only mineral that is isotopically lighter than silicate melt at equilibrium[14], fractional crystallization of plagioclase during plutonic cooling at shallow depths (which would not affect the inherited Dy/Yb signatures) provides the most obvious means of increasing melt δ⁴⁴Ca to values heavier than BSE (see Supplementary Note 5 and Supplementary Fig. 5), as observed in three of our TTG samples. Plagioclase fractionation must have also occurred in many of the other samples, suggesting that measured δ⁴⁴Ca likely represent maximum values for parental TTG melts, prior to crystallization, and that our geothermal gradient predictions generally represent upper estimates (Supplementary Note 5).

**Assimilation of carbonate sediments.** Of the five samples that lie below our model predictions for 500 °C/GPa, two have positive

show the evolving δ⁴⁴Ca of silicate melts during progressive heating along three different geothermal gradients for two end-member scenarios (closed-system equilibrium vs. garnet fractionation, Supplementary Data 4). The overall trends are dominated by the competition between increasing temperature (decreasing isotopic fractionation) and increasing pressure (increasing

Eu/Eu* anomalies that can be explained through plagioclase accumulation (Supplementary Note 5) and one deviates only slightly from the predictions and could suggest formation along moderately lower geothermal gradients. The two NSB samples (~3.8 Ga, Fig. 1a), however, have unusually low $\delta^{44}Ca$ that cannot be explained by equilibrium magmatic processes. This implies that these samples record either (i) kinetic isotope fractionations or (ii) incorporation of isotopically distinct Ca-rich materials, not recorded by other samples. In volcanic systems, large kinetic isotope effects can result from Ca diffusion during rapid crystal growth[21]. This mechanism cannot explain the composition of NSB samples, however, because a negative shift in bulk magma $\delta^{44}Ca$ would require rapid growth and removal of minerals where Ca is strongly incompatible, which (by definition) would have little effect on the Ca budget. The TTG samples also (i) lack compositional banding, (ii) were collected far from lithological contacts, (iii) have similar initial Hf isotopic compositions in zircons and their host rocks[11], and (iv) have no apparent correlations between $\delta^{44}Ca$ and fluid mobile/immobile trace-element ratios (Supplementary Fig. 6), suggesting that significant changes in bulk-rock chemistry associated with post-emplacement metamorphism/metasomatism did not take place. Although they are generally similar to other TTGs in terms of major/trace-element chemistry and Si isotopic compositions (Supplementary Fig. 7), the NSB samples have much higher peraluminosity [denoted "A/CNK," defined as molar $Al_2O_3/(CaO+Na_2O+K_2O)$] and oxygen isotope ratios ($\delta^{18}O$), which are typically attributed to assimilation of metasediments[22] (Fig. 3a). Thus, a diffusion-based fractionation mechanism, though possible, would imply that the high peraluminosity and elevated $\delta^{18}O$ in NSB samples are merely coincidental. The most parsimonious explanation for low $\delta^{44}Ca$ in these samples therefore is the incorporation of metasedimentary materials [e.g., carbonated ("marly") sediments] with high A/CNK, high $\delta^{18}O$, and low $\delta^{44}Ca$ values.

Calcium isotopes have been used as a proxy for recycled carbonates in a number of magmatic systems; however, many previous studies assume low $\delta^{44}Ca$ for carbonates (e.g., −1‰) that are not representative of global averages (Supplementary Note 6a). Average Phanerozoic carbonates have $\delta^{44}Ca$ only slightly lower than BSE [−0.35‰[23]], which implies that NSB samples cannot be explained through incorporation of carbonates with average $\delta^{44}Ca$. The high internal heat budget of early Earth[4], however, may have promoted hydrothermally buffered (BSE-like) seawater $\delta^{44}Ca$ (Supplementary Note 6b), in contrast with modern seawater, which has evolved to high $\delta^{44}Ca$ (+0.9‰) over the Phanerozoic[24]. Thus, with the same average fractionation between seawater and carbonates as today (Supplementary Note 6b), BSE-like seawater could have precipitated carbonates with an average $\delta^{44}Ca$ of −1.25‰ (Fig. 3b). Using a three endmember mixing model (average TTG, average shale, and average carbonate with $\delta^{44}Ca = -1.25‰$) shown in Fig. 3a, c, we find that NSB samples are best explained through incorporation of 30–50 wt% carbonated metasediments (shale and/or hyaloclastite containing 5–20 wt% carbonates, see "Methods"). Altered oceanic crust could also potentially serve as a source of carbonates in TTG protoliths, but given that these carbonates typically have higher $\delta^{44}Ca$ values[25], it is unlikely that they provide enough isotopic leverage to explain the NSB samples (Supplementary Note 6a).

As depicted in Fig. 4, our data provide further evidence for the incorporation of surficial materials (carbonates and shales) into TTG sources, while Si isotope data for the same samples suggest incorporation of chert[9,10]. Thus, several types of sediment [often found together in accretionary wedges (Supplementary Note 7)] could undergo subduction and melting in the early Eoarchean. The ubiquity of Si isotope signatures indicative of chert incorporation in TTGs, however, contrasts with the limited

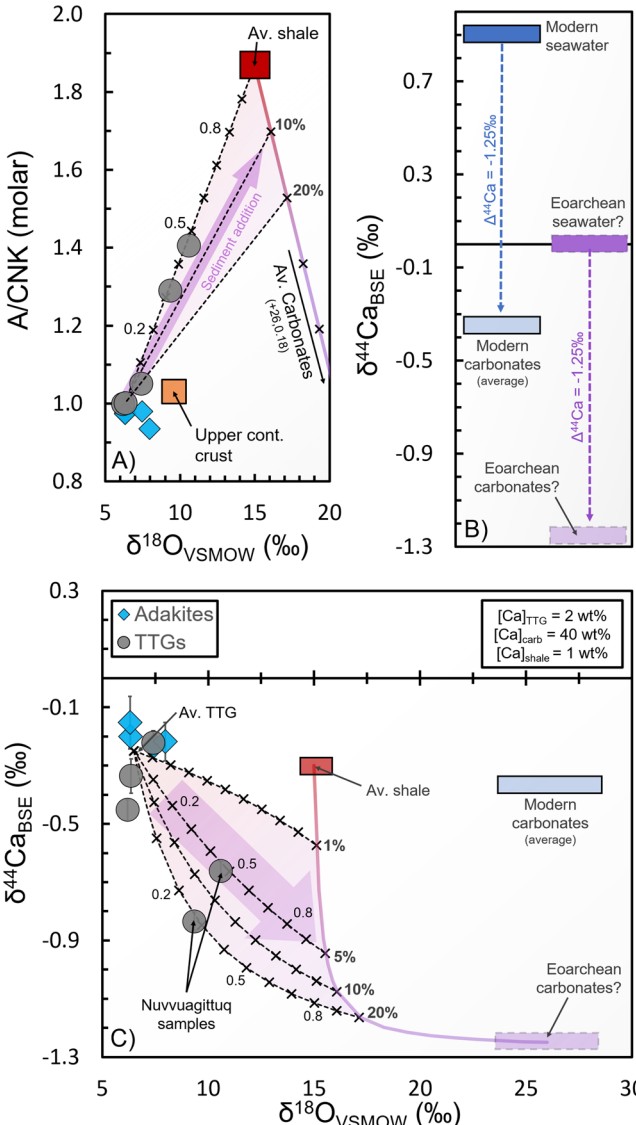

**Fig. 3 Three end-member mixing models (see "Methods" for details) for sediment assimilation in >3.8 Ga Nuvvuagittuq Supracrustal Belt (NSB) samples. A** Peraluminosity index (A/CNK) vs. $\delta^{18}O$ ($n = 9$). **B** Hypothesized Eoarchean carbonate $\delta^{44}Ca$ based on BSE-like seawater and average modern $\Delta^{44}Ca_{carb-sw}$ ($\delta^{44}Ca_{carb} - \delta^{44}Ca_{sw}$, Supplementary Note 6). **C** $\delta^{44}Ca$ vs. $\delta^{18}O$ ($n = 9$). These results suggest that NSB samples incorporated 30–50% shale (with 5–20% carbonates). Error bars represent 2SE uncertainties on our Ca isotope measurements. Av. TTG average TTG, upper cont. crust upper continental crust, TTG tonalite–trondjhemite–granodiorite suite.

number of samples that require assimilation of carbonates, potentially indicating that carbonates were geographically limited, difficult to incorporate into melts during subduction, and/or generally had $\delta^{44}Ca$ signatures similar to average TTGs (e.g., −0.3‰, which would represent slow/equilibrium precipitation from BSE-like seawater at ~25 °C, Supplementary Note 6). Thus, we cannot rule out that TTGs other than the NSB samples also incorporated carbonates. Given that their $\delta^{44}Ca$ values can be reproduced through magmatic processes, however, carbonates are not necessary to explain the bulk of our data.

Our results constrain the apparent geothermal gradients along which oceanic crust (and metasediments) melted to produce TTG magmas, providing further evidence for the early operation of subduction-driven plate tectonics (Fig. 4) and contrasting with

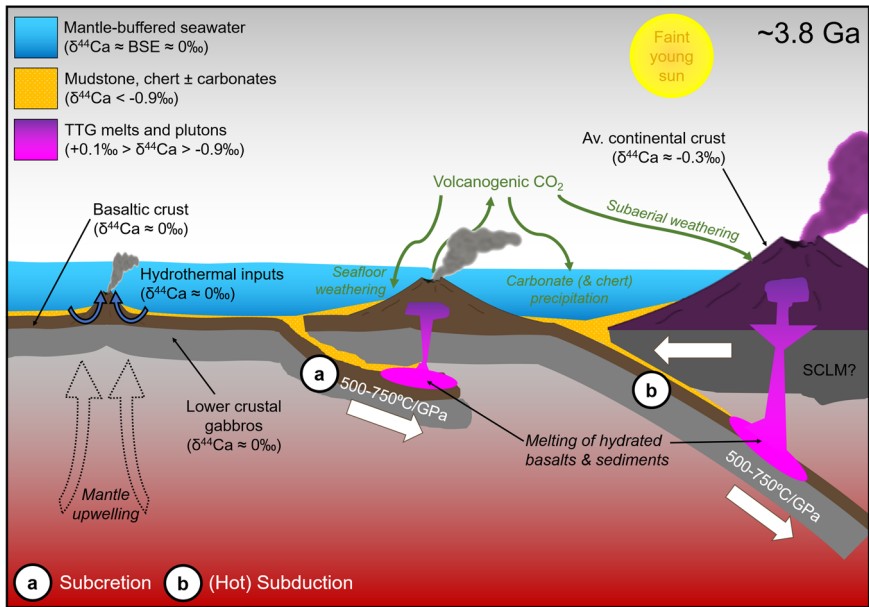

**Fig. 4 Schematic diagram of geodynamic settings supported by our TTG data [e.g. subcretion (a), localized roll-back (not shown)[7], and/or modern-style subduction (b)].** Apparent geothermal gradients (500–750 °C/GPa) are similar to those reported for modern adakites[5,6] and hot subduction-zone assemblages[16]. Vigorous hydrothermal circulation promotes mantle-buffered seawater that can lead to isotopically light $\delta^{44}Ca$ (<−0.9‰) in precipitated carbonates (see Supplementary Note 6), providing a sink for high atmospheric $CO_2$[30]. Carbonates [along with cherts[9] ± shales] can then be subducted and incorporated into TTG sources. Av. continental crust average continental crust, BSE bulk-silicate Earth.

estimates relying on the appearance of blueschists or of paired metamorphic belts in the rock record[26]. Although we are analyzing the products of melting, as opposed to the metamorphic residues, our results suggest that relatively low d$T$/d$P$ conditions indicative of subduction may be underrepresented in the ancient metamorphic record (Supplementary Fig. 8), in agreement with previous predictions[27]. The data also provide independent evidence for the presence of carbonated sediments on the ocean floor in the early Eoarchean (>3.8 Ga), predating the oldest carbonate units preserved in the rock record[28] and suggesting that carbonates from earlier periods may be underrepresented. This implies that the silicate–carbonate cycle was in place by ~3.8 Ga and provided a necessary sink for high levels of volcanic $CO_2$ outgassing [which would have otherwise led to a runaway greenhouse effect on Earth[29]]. The results thus not only provide support for models that require elevated atmospheric p$CO_2$ to compensate for the faint young sun[30] but also have significant implications for continental emergence/weathering through time[31,32]. Although our data do not definitively prove the existence of a global network of interconnected plate boundaries (as required by some definitions of "plate tectonics"), we show that subduction events occurred repeatedly throughout the Archean, in agreement with a growing body of evidence suggesting that plate-tectonic processes started prior to 3.5 Ga[1,6,9,10,20,31–34].

## Methods

**Chemical and isotopic analyses**. In order to perform chemical and isotopic analyses relevant to this study, aliquots of powdered rock sample (~20–50 mg) are first dissolved in 2:1 mixtures of concentrated HF + 6 M HCl and refluxed at 130 °C for 1 week. The samples are then evaporated to dryness and fully redissolved in 4 M $HNO_3$ (also refluxed at 130 °C) for subsequent Ca isotope and major/trace-element analyses. Sample descriptions are available in Supplementary Note 1 and Supplementary Data 2.

*Ca isotope analyses*. Calcium (~50 μg) is separated from aliquots of the dissolved samples and standards using established column chemistry methods adapted from UC-Berkeley[21,35–38] and modified for multi-collector inductively coupled plasma

mass spectrometry (MC-ICP-MS) analyses[39]. Eichrom® DGA resin is thoroughly washed by alternating between ultrapure water and 4 M $HNO_3$ several times overnight before loading into home-made Teflon columns with ~2 mL reservoirs. The resin (~250 μL) is loaded into the columns and washed with 2 mL $H_2O$, 2 mL 4 M $HNO_3$, 2 mL $H_2O$, 1 mL 6 M HCl, 2 mL $H_2O$, and 2 mL 4 M $HNO_3$. Samples are then loaded onto columns in ~1 mL of 4 M $HNO_3$, and a total of 6.8 mL of 4 M $HNO_3$ (in progressively larger batches) is used to elute matrix elements. The Ca is then collected using 3.5 mL of ultrapure water (500, 1000, 2000 μL), evaporated to dryness, and then redissolved with one drop of ~50% $H_2O_2$ and concentrated $HNO_3$ (allowed to sit overnight to destroy any leached organics or resin that may have passed through the column frits). The samples are evaporated to dryness and redissolved in 1 mL of 4 M $HNO_3$ to repeat the entire process with fresh resin, ensuring the effective removal of matrix elements with isobaric interferences (e.g., Sr). After the second pass through the columns, the purified Ca from samples and standards is dissolved in 5 mL 0.1 M $HNO_3$ (to make ~10 parts per million (ppm) Ca solutions) for isotopic analysis.

Calcium isotope compositions ($^{44}Ca/^{42}Ca$ and $^{44}Ca/^{43}Ca$) are measured via MC-ICP-MS using sample-standard bracketing on a Thermo Fisher Neptune instrument at IPGP. A majority of our measurements use SRM915b as a bracketing standard due to greater availability. To allow for easier comparison with published data and to best illustrate complementarity between terrestrial reservoirs, $\delta^{44/42}Ca$ values are converted to $\delta^{44}Ca$ by multiplying by 2.05 (the kinetic mass law) and are reported relative to BSE [with a recommended value of +0.95‰ relative to SRM915a[14], which is +0.25‰ relative to SRM915b in this study]. As $^{40}Ca$ is not directly measured, the use of MC-ICP-MS for stable Ca isotope analyses precludes the need for radiogenic $^{40}Ca$ corrections, which could otherwise lead to significant variations in Archean-aged samples analyzed by thermal-ionization mass spectrometry[35,36,40].

Samples are introduced into the mass spectrometer using an Apex desolvating nebulizer attached to an autosampler system with a probe aspiration rate of ~100 μL/min. A rinse time of 150 s (using 0.1 M $HNO_3$) effectively washes residual sample out of the system between measurements. High-resolution slits allow for a mass resolution ($M/\Delta M$) of ~6350 and accurate measurement of $^{42}Ca$, $^{43}Ca$, and $^{44}Ca$ signals. Each analysis consists of 20 cycles with an integration time of 8.389 s/cycle, and each sample is analyzed a minimum of five times. The signals for $^{44}Ca$ are at least ~4 volts, and all analyses give mass-dependent relationships between $^{44}Ca/^{43}Ca$ and $^{44}Ca/^{42}Ca$, confirming the effective removal of matrix elements with potential isobaric interferences.

Ca isotope data for samples and standards in this study are reported in Supplementary Data 1, with individual analyses reported in Supplementary Data 3. Repeated analyses of SRM915b (bracketed by SRM915a) yield a $\delta^{44}Ca$ of +0.70 ± 0.02‰ (2SE, $n = 10$), in agreement with previous studies[41] and the recommended value[14]. Several of our TTG samples are additionally run against SRM915a, which gives a similar offset between SRM915a and SRM915b [+ 0.69 ± 0.02‰ (2SE, $n = 3$)] to that observed when they are measured directly against each other. USGS

reference materials W2-a (dolerite) and AGV-2 (andesite) are also measured alongside our samples and yield values of $+0.79 \pm 0.03‰$ (2SE, $n = 19$) and $+0.72 \pm 0.02‰$ (2SE, $n = 9$), relative to SRM915a, respectively, in agreement with previous studies[36,41–44]. This suggests that our analytical methods are robust and suitable for the silicate rocks targeted in this study.

*Major and trace-element analyses.* Element concentration measurements are conducted using an Agilent 7900 quadrupole inductively coupled plasma mass spectrometer (Q-ICP-MS) at IPGP. Aliquots of ~20–100 µL (representing 0.5 mg of rock) are taken from each dissolved sample and diluted prior to analysis in order to produce concentrations within the calibrated range of the instrument. The results are calibrated against multiple in-house standards, with calibration standards intermittently checked every ten samples to ensure accurate standardization of the analyses. Calibration standards and the internal element standards Sc and In are used to monitor and correct for drift and matrix effects. Major element concentrations are reported in wt% and trace-elements are reported in ppm (Supplementary Data 1). Our elemental data for USGS standards AGV-2 and W2-a agree with accepted values from the GeoREM database[45], and TTG samples agree well with previously reported measurements[46–50]. Reproducibility for all elements is better than 5% (relative standard deviation).

**Ca isotope phase-equilibrium modeling**. In order to predict the stable Ca isotope composition of melts produced during progressive heating along different geothermal gradients, we incorporate ab initio estimates for equilibrium Ca isotope fractionation into the phase equilibrium models described above. This represents a significant improvement over previous modeling attempts where temperatures, pressures, and/or mineral proportions/compositions are fixed[15,51] and gives a more realistic picture of Ca isotopic fractionation during melt production in evolving magmatic systems. Phase equilibrium modeling methods are described in Supplementary Note 2 and are based on those of ref. [12]. Reduced partition function ratios (RPFRs) used in our Ca isotope models are discussed in Supplementary Note 3 and compiled in Supplementary Table 2.

*Mineral–melt Ca isotope fractionation.* To account for decreasing isotopic fractionation with increasing temperatures, the RPFR for each phase is recalculated at each temperature (where $T$ is in Kelvin) according to:

$$\ln\beta(T) = \ln\beta(1000\,\text{K})\frac{10^6}{T^2} \quad (1)$$

The Ca isotope fractionation factor between two phases $a$ and $b$, $(\alpha_{a-b})$, at a given temperature can then be calculated as the ratio of their temperature-dependent RPFRs:

$$\alpha_{a-b}(T) = \frac{\beta_a(T)}{\beta_b(T)} \quad (2)$$

In the case of determining fractionation between melt and a set of minerals with changing proportions/compositions, the effective RPFR for the bulk solid phase is recalculated at each step by combining the RPFRs for each mineral and weighing them by their relative contributions to the total Ca budget of the solids, where $X^i_{\text{Ca}}(P,T,x)$ is the fraction of total Ca in phase $i$ (which varies with pressure, temperature, and bulk composition) and comes from the phase equilibrium modeling results.

$$\alpha_{\text{solids}-\text{melt}}(P,T,x) = \frac{\sum^i_{\text{solids}}[X^i_{\text{Ca}}(P,T,x)\beta_i(T)]/[1 - X^{\text{melt}}_{\text{Ca}}(P,T,x)]}{\beta_{\text{melt}}(T)} \quad (3)$$

*Closed-system equilibrium calculations.* In order to model the isotopic evolution of Ca during closed-system melting of a starting material with BSE $\delta^{44}$Ca, we start from the mass-conservation equation for equilibrium trace-element distributions between solids and melt in a closed system:

$$\frac{C_l}{C_o} = \frac{1}{D_{\text{Ca}} + f(1 - D_{\text{Ca}})} \quad (4)$$

where $C_l$ is the concentration of Ca in the melt, $C_o$ is the Ca concentration in the bulk system, $f$ is the melt fraction, and $D_{\text{Ca}}$ is the Ca distribution coefficient ($D_{\text{Ca}} = [\text{Ca}]_{\text{solids}}/[\text{Ca}]_{\text{melt}}$). Treating $^{40}$Ca and $^{44}$Ca separately and using the trace-abundance approximation[52–54], where the distribution coefficient for $^{40}$Ca (~97% of all Ca) is assumed to be equal to that of bulk Ca and the distribution coefficient for $^{44}$Ca is equal to $D_{\text{Ca}}$ multiplied by $\alpha_{\text{solids}-\text{melt}}$, we arrive at:

$$\left(\frac{^{44}\text{Ca}}{^{40}\text{Ca}}\right)_{\text{melt}} = \left(\frac{^{44}\text{Ca}}{^{40}\text{Ca}}\right)_o$$
$$\left[\frac{1 + f[1/D_{\text{Ca}}(P,T,x) - 1]}{\alpha_{\text{solids}-\text{melt}}(P,T,x) + f[1/D_{\text{Ca}}(P,T,x) - \alpha_{\text{solids}-\text{melt}}(P,T,x)]}\right] \quad (5)$$

where $D_{\text{Ca}}(P,T,x)$ is calculated from the phase-equilibrium model results at each step. The $\delta^{44}$Ca of silicate melt can then be calculated:

$$\delta^{44}\text{Ca}_{\text{melt}} = \left[\frac{(^{44}\text{Ca}/^{40}\text{Ca})_{\text{melt}}}{(^{44}\text{Ca}/^{40}\text{Ca})_{\text{BSE}}}\right] - 1 \quad (6)$$

In the closed-system scenario, we assume that $\delta^{44}$Ca$_{\text{bulk}} = 0‰$. Comparing the

results for Eq. 5 with a commonly used approximation of the form:

$$\delta^{44}\text{Ca}_{\text{melt}} = \delta^{44}\text{Ca}_{\text{bulk}} - [1 - X^{\text{melt}}_{\text{Ca}}(P,T,x)]\ln\alpha_{\text{solids}-\text{melt}}(P,T,x) \quad (7)$$

We find that the results agree within <0.1 ppm (equivalent to <0.0001‰), which is negligible compared to typical analytical uncertainties (~0.1‰), and so we use the simpler equation (Eq. 7) in our models.

*Garnet fractionation calculations.* In order to simulate the progressive isolation of garnet cores from the rest of the bulk system, we evolve the bulk chemical and isotopic compositions of the system such that 5/6 of the garnet is removed every time 5 mol% garnet is reached [equivalent to ~6 wt%, depending on the solid–solution compositions[12]]. Afterwards, 1/6 of the garnet (e.g., the reactive rims) remains in the system and equilibrates chemically and isotopically with the other mineral phases and melt. The isotopic evolution of the system is thus calculated in an iterative fashion according to:

$$\delta^{44}\text{Ca}_{\text{Bulk}}(n+1) = \begin{cases} \delta^{44}\text{Ca}_{\text{nongrt}}(n)\left[1 - X^{\text{grt}}_{\text{Ca}}(n)\right] + \delta^{44}\text{Ca}_{\text{grt}}(n)X^{\text{grt}}_{\text{Ca}}(n) \text{ if } \frac{1}{6}n \neq \mathbb{N}_1 \\ \delta^{44}\text{Ca}_{\text{nongrt}}(n)\left[1 - \frac{1}{6}X^{\text{grt}}_{\text{Ca}}(n)\right] + \delta^{44}\text{Ca}_{\text{grt}}(n)\frac{1}{6}X^{\text{grt}}_{\text{Ca}}(n) \text{ if } \frac{1}{6}n = \mathbb{N}_1 \end{cases}$$
$$(8)$$

where $n$ is the number of steps (1 mol% of garnet growth, starting at $n = 0$) in the iterative calculation, $\delta^{44}\text{Ca}_{\text{nongrt}}(n)$ and $\delta^{44}\text{Ca}_{\text{grt}}(n)$ are the Ca isotopic compositions of all non-garnet phases (including melt) and of garnet, respectively (in the previous step), $X^i_{\text{Ca}}(n)$ comes from the phase-equilibrium modeling results, and $\mathbb{N}_1$ indicates natural numbers (beginning with 1). Note that removing larger batches of garnet (e.g., 10%) would decrease the $\delta^{44}$Ca effects predicted in our models, while smaller batches (e.g., 1%) would increase the predicted effects and approach those that would be predicted by a Rayleigh fractionation model. For simplicity, we only show the dependence of each above variable on $n$; however, we note that each of these variables also depends on pressure, temperature, and bulk system composition ($P,T,x$), as in the previous closed-system calculations (Eqs. 1–7). The Ca isotopic compositions of non-garnet (including melt) and garnet phases are calculated:

$$\delta^{44}\text{Ca}_{\text{nongrt}}(n) = \delta^{44}\text{Ca}_{\text{bulk}}(n) + \ln\alpha_{\text{nongrt}-\text{grt}}(n)[X^{\text{grt}}_{\text{Ca}}(n)] \quad (9)$$

$$\delta^{44}\text{Ca}_{\text{grt}}(n) = \delta^{44}\text{Ca}_{\text{bulk}}(n) - \ln\alpha_{\text{nongrt}-\text{grt}}(n)[1 - X^{\text{grt}}_{\text{Ca}}(n)] \quad (10)$$

where we assume that $\delta^{44}\text{Ca}_{\text{bulk}}(n = 0)$ is equal to BSE (=0‰) and where the Ca fraction of each phase $i$ [$X^i_{\text{Ca}}(n)$] comes from the phase-equilibrium modeling results. The fractionation factor between garnet and non-garnet phases is calculated:

$$\alpha_{\text{nongrt}-\text{grt}}(n) = \frac{\sum^i_{\text{nongrt}}[X^i_{\text{Ca}}(n)\beta_i(T)]/[1 - X^{\text{grt}}_{\text{Ca}}(n)]}{\beta_{\text{grt}}(T)} \quad (11)$$

After calculating the bulk composition of the system at each step of progressive garnet growth and sequestration (Eqs. 8–11), the $\delta^{44}$Ca of the melt can then be calculated in the same fashion as for the closed-system equilibrium case (Eq. 7) but with each variable depending on the evolving $\delta^{44}\text{Ca}_{\text{bulk}}(n)$ and bulk chemical composition of the system as garnet is removed (Eq. 12).

$$\delta^{44}\text{Ca}_{\text{melt}} = \delta^{44}\text{Ca}_{\text{bulk}}(n) - \left[1 - X^{\text{melt}}_{\text{Ca}}(n)\right]\ln\alpha_{\text{solids}-\text{melt}}(n) \quad (12)$$

**Mixing model parameters**. In order to account for the three most important parameters indicative of sediment assimilation in NSB TTGs (high $\delta^{18}$O, high A/CNK, and low $\delta^{44}$Ca) we chose to perform three end-member mixing calculations in A/CNK vs. $\delta^{18}$O and $\delta^{44}$Ca vs. $\delta^{18}$O parameter space, with the results shown in Fig. 3 of the main text. We use (i) average shale[22,55] [$\delta^{18}$O = +15‰, A/CNK = 1.87, and [Ca] = 1 wt%] with $\delta^{44}$Ca = −0.3‰ [the average value from shale standards SGR-1 and SBC-1[39,44,56,57]], (ii) average TTG ($\delta^{18}$O = +6.5‰, A/CNK = 1.02, and [Ca] = 2 wt%) with $\delta^{44}$Ca = −0.25‰ (this study), and (iii) average carbonate [$\delta^{18}$O = +26‰, A/CNK = 0.18, and [Ca] = 40 wt%[22]] with $\delta^{44}$Ca = −1.25‰ (corresponding to precipitation from BSE-like seawater). The A/CNK of average carbonate comes from a compilation of data for Phanerozoic limestones with available $Al_2O_3$, CaO, $Na_2O$, and $K_2O$ concentrations ($n = 908$) from the Earthchem database (http://www.earthchem.org/portal downloaded June 26, 2020, Supplementary Data 5). We note that assimilation of average continental crust cannot lead to high enough A/CNK or $\delta^{18}$O values [shown in Fig. 3 of the main text[22,58]] to explain the composition of our NSB samples. We assume that (i) all three end-members have equivalent [O] concentrations, (ii) A/CNK mixes linearly, and that (iii) $\delta^{44}$Ca is dependent on [Ca] and thus produces curved mixing lines in $\delta^{44}$Ca vs. [Ca] parameter space (cf. straight mixing lines in ref. [59]). For NSB sample INO5003, we find that an assimilated sediment fraction (80% shale with 20% carbonates) of ~0.3 explains the A/CNK, $\delta^{18}$O, and $\delta^{44}$Ca values, while sample INO5012 requires a sediment fraction of ~0.5 (95% shale with 5% carbonates). The mixing calculations yield final [Ca] concentrations for INO5003 and INO5012 of 4.0 and 2.5 wt%, respectively, which is moderately (but not grossly) higher than measured by Q-ICP-MS (Supplementary Data 1), potentially suggesting that the average [Ca] used in our mixing models (for the TTG and/or shale end-members) may be slightly high. Lower [Ca] estimates for TTG and shale end-members would mainly serve to lower [Ca] in the resulting

mixtures but would not lead to significant differences in the estimated sediment fractions (due to the overwhelming influence of carbonate [Ca] on $\delta^{44}$Ca). The similar estimates for sediment fractions in both A/CNK vs. $\delta^{18}$O and $\delta^{44}$Ca vs. $\delta^{18}$O space, however, suggest that our mixing models are relatively robust and that the estimated parameters (e.g., $\delta^{44}$Ca of Eoarchean carbonates) are reasonable.

## Data availability

All data are available in the main text, Supplementary Information, and Supplementary Data 1–5.

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

## Acknowledgements
M.A.A. is grateful for discussions regarding the data and modeling contained in this work and would like to thank James Farquhar, Donald J. DePaolo, C. Brenhin Keller, and Olivier Bachmann for their insight. We also thank Pascale Louvat, Barthelemy Julien, and Pierre Burckel for their help with mass spectrometry and trace-element analyses. This article is dedicated to the memory of Professor Peter L. Antonelli. F.M. acknowledges funding from the European Research Council under the H2020 framework program/ERC grant agreement #637503 (Pristine). Parts of this work were also supported by IPGP multidisciplinary program PARI, Region Île-de-France SESAME Grants no. 12015908 and EX047016, and IdEx Université de Paris grant ANR-18-IDEX-0001. M.A. A. acknowledges support from an ETH postdoctoral fellowship (19-2 FEL-33). C.Y. would like to acknowledge funding from NSERC Discovery Grant. J.K. was funded by an NSERC Vanier Scholarship. M.G. acknowledges funding from the French National Research Agency (ANR) through grants ANR-10-LABX-0006 (ClerVolc) and ANR-17-CE31-0021 (Zircontinents). This is laboratory of Excellence ClerVolc contribution no. 457.

## Author contributions
M.A.A. and F.M. designed research; M.A.A. prepared and analyzed the samples; M.A.A., J.K., and C.Y. developed isotopic phase-equilibrium models; M.G. provided samples; M.A.A. interpreted the data and wrote the paper with input from C.Y., J.K., M.G., T.M., and F.M.

## Competing interests
The authors declare no competing interests.
