## [Peer Review File · Nature Communications]

REVIEWER COMMENTS

Reviewer #2 (Remarks to the Author):

Review of: “Calcium isotope evidence for early Archaean carbonates and plate tectonics”

This manuscript is provocative and showcases a novel use of an under-appreciated stable isotope system to reveal the geological evolution of a poorly understood terrane – the Nuvvuagittuq supracrustal belt, Canada. The study thus effectively combines real-world petrology with isotope geochemistry and modelling. I had some small queries about the preferred open vs. closed petrological modelling results and the interpreted geothermal gradient for TTG magma genesis, but my confusion may simply be due to the need to keep the word count down in the main text for this journal. A short expansion on these points would help me and presumably other readers to have a clear idea of the tectonic model put forwards, although Figure 4 is very effective in summarizing this. In short, I think that this is an excellent study and the results have a potentially high impact on our understanding of early Earth geodynamics, so it is well suited for publication in *Nature Communications*. Please do not mistake my brief review comments below for a lack of interest in the subject or a lack of attention while reading it – the work is well structured and edited, the data seem reliable, and the interpretations are defensible, so I had relatively little to say.

Title: I rarely comment on titles, but I note that you explicitly mention evidence for plate tectonics, whilst being more circumspect (rightly so) in your conclusions (lines 190-193). Should the title be amended accordingly?

Line 42: I would argue that this needs rewording. First, I assume you mean a combined TTG suite instead of the individual components, which of course may form individually by a variety of processes. Secondly, what is the “modern Earth”? Post-initiation of plate tectonics? The authors know this – I am quite sure – but TTGs likely formed in the early stages of a mobile lid regime when subduction was warm and slabs dipped at shallow angles (see geodynamic regime diagram below from Palin et al., 2020; *Earth-Science Reviews*). So, is the “modern Earth” defined by cold, steep, and deep subduction only? If so, when did *that* start?

[redacted]

Line 102: Here (i.e. in Fig 2B), it seems quite fanciful to say that the data fit best *between* 500 °C/GPa and 750 °C/GPa, when in fact they look like they fit the 500 °C/GPa curve very well by themselves. More data points seem to lie on what would be the cold-temperature side of it than the high-temperature side. And it’s also worth pointing out that the dotted line representing open system (fractionation) of garnet during metamorphism of the DAT appears to veer away from the measured TTG data considerably above ~1.5 GPa (so, 750 °C at this gradient), but the closed-system (equilibrium) scenario fits the TTG data quite well across

the entire pressure and temperature range. Yet, you then show on Fig. 2 that TTG magma genesis occurred at temperatures above 700 °C, so is this the scenario you're championing? Closed-system melting? When does the final melt leave the system to form the plutons in the overlying crust? Is there a physical reason that it would stay in place after it exceeds the well-known melt connectivity threshold of ~7 vol. %? I'm purely thinking of the physical implications of your modelling results, here. Finally, how does this trace element analysis tie in to later plagioclase fractionation within the crustal magma chamber (lines 119-122) – this process wouldn't change the already-inherited Dy/Yb ratio, I assume? I realize that words are at a premium for *Nature* journals, but I think interpretation of the physical processes of melt formation and loss can be expanded upon slightly.

Lines 113-114: I hate to bring this up, but Palin et al. (2016), who published these calculations and results a year earlier, should replace this reference (Johnson et al., 2017). The study referenced here performed almost identical work and – surprisingly – did not cite us (cf. the corrigendum written by Johnson et al. in response to a query on this issue made to the journal editor: www.nature.com/articles/nature22385). It is also worth noting that the bulk composition used by Johnson et al. (2017) is highly atypical of Archean depleted basalts (very low Mg#), as also acknowledged in their corrigendum, rendering their results of limited value for broad-scale interpretation of Archean processes.

- Palin, R.M., White, R.W. and Green, E.C.R., 2016. Partial melting of metabasic rocks and the generation of tonalitic–trondhjemitic–granodioritic (TTG) crust in the Archaean: Constraints from phase equilibrium modelling. *Precambrian Research*, 287, 73-90.

Lines 114-116: All Archean TTGs or just the ones you are studying?

Line 153: This should be “marly”, not “marley”, presumably?

Lines 167-169: This seems a bit speculative, although the results of your analysis for the Nuvvuagittuq samples depends on it being true. Is there support for having the same average fractionation between sediment and seawater at this early point in Earth history?

Lines 169-171: I didn't catch whether this was weight % or volume %, but either way, you're suggesting that up to half of the TTG magma analysed is assimilated shale plus carbonate sediment? That is a lot of sediment to transport down a subduction channel when most of the seafloor material is scraped away and forms accretionary prisms above the trench. Are there data to support this? Isn't it also plausible that the sediment would dewater (or melt?) and transform into a metasediment at some point during burial *before* the subducted mafic crust melts?

Line 191-192: This is an important caveat to note (one instance of subduction \neq global plate tectonics), but how do your data show that “*subduction occurred continuously throughout the Archaean*”? Sorry if I missed something here.

Lines 495-513: There are swaths of the Methods section that appear to be further discussion (i.e. not so much outlining the work that was done). This is a prime example and gives the worrying impression that some critical information or justification for the calculations performed is kept separate from the main text. I'd think about going through this Methods section to eliminate arguments for why you did what you did, and just stick to what you did (and how you did it)!

Supplementary Information

Lines 79-82: Can the authors explain why 5 mol % was used as a trigger point for performing fractionation? It seems arbitrary. Aren't components sequestered into garnet continuously during growth, and doesn't garnet grow discontinuously during metamorphism? Would the results differ if a step of 1 mol % or 10 mol % was used?

Lines 127-243: This is another example of where there is extended discussion (i.e. line 134: "*In the following section, we provide a short review of modern geophysical observations for oceanic plateaus and adakitic magmatism*"). Although interesting, it seems like overkill when past studies can be cited to support the choice of geotherm that characterizes different environments. If I put this another way, there are around 1600 words in this section alone (section 4 of the S.I.); nearly two thirds of the manuscript itself.

Lines 194-196: Hornblende is a candidate, though, and many studies have suggested that TTGs form due to melting of amphibolite or garnet amphibolite. Thus, you don't need trans-crustal hydrothermal systems if you're breaking down a hornblende-bearing rock and releasing structurally bound water.

Extended Data Fig. 2: Are the lower panels showing the Ca weight % content of the minerals in question or the weight % of minerals that just contain Ca?

Richard Palin
Associate Professor of Petrology, University of Oxford

Reviewer #3 (Remarks to the Author):

The authors measured Ca isotopes in Archean granitoids and used the data to obtain some information about their origin and about the geodynamic regime that operated at this time. These ideas are novel and will they be of interest to many others in the community and the wider field. The new data seem to be of high quality and well worth publishing. However, I find many aspects of the interpretation unconvincing.

1. In the introduction the authors equate subduction and plate tectonics; even though I think it is difficult to have one without the other, many others argue that demonstrating subduction does not imply the existence of rigid plates which, for them, are an essential component of plate tectonics.
2. The correlations between $\delta^{44}\text{Ca}$ and the trace-element ratios that are used by the authors to infer PT trajectories are really rather poor. Three samples fall well above the line in Fig 2b; this feature is attributed to plagioclase fractionation. Five others fall below the line, which might imply low temperatures. For the Nuvvuagittuq samples, the low values are attributed to the presence of carbonate; the low- $\delta^{44}\text{Ca}$ of the others are not discussed. The position of the remaining 9 samples with intermediate $\delta^{44}\text{Ca}$ is shown to correspond to the modelled 500-750°C/GPa trajectory which is compared to that of a hot modern subduction zone. This is then taken as an argument that subduction operated at the time these granitoids were formed. This conclusion is not unreasonable (in my opinion, though others might disagree) but the line of argument seems rather tenuous.
3. To explain the low $\delta^{44}\text{Ca}$ of the Nuvvuagittuq samples, the authors propose that the source contained carbonate sediments. For this to fit, they follow a rather tortuous path. Modern carbonates have only slightly negative $\delta^{44}\text{Ca}$ like those of the majority of the TTGs so the presence of this material can't explain the very low values of the Nuvvuagittuq samples. The authors therefore propose that "the high internal heat-budget of early Earth ... promoted hydrothermally buffered Eoarchean seawater" to near BSE values, which would have led to strongly negative values in Eoarchean carbonate. This is not unreasonable, but it raises the question why this process did not operate during the formation of the other TTG, which have only slightly younger ages. Why was carbonate incorporated into the source of the Nuvvuagittuq samples but not that of the slightly younger samples, which formed at a time when the mantle was just as hot?
4. The Nuvvuagittuq samples have elevated $A/\text{CNK} = \text{molar Al}_2\text{O}_3/(\text{CaO}+\text{Na}_2\text{O}+\text{K}_2\text{O})$ which is attributed to the presence of clay-rich sediment. However, a high A/CNK is difficult to reconcile with the presence of Ca-rich carbonate, which would decrease A/CNK . The authors claim to have found a recipe that works but this all seems rather contrived.
5. A role for carbonate in the Eoarchean granitoids is itself problematic, given that carbonate sediments are virtually absent from all Archean sequences, most probably due to the absence of shell-forming organisms and very different seawater compositions at that time.

An alternative explanation that might be explored is that terrestrial sediments are not involved; rather, the carbonate could have been a secondary phase in altered oceanic crust. Volcanic rocks in Archean greenstone belts commonly contain secondary carbonate. A distinctive feature of the Nuvvuagittuq rocks is that they formed at a time when very little continent was around and those continents that were around were largely flooded. Terrestrial sediment would have been virtually absent. The distinctive geochemistry of these rocks may reflect the presence in their source of carbonatized oceanic crust and deep-sea clay.

Nicholas Arndt
Grenoble, December 2020

REVIEW RESPONSES

Reviewer #2 (Remarks to the Author):

This manuscript is provocative and showcases a novel use of an under-appreciated stable isotope system to reveal the geological evolution of a poorly understood terrane – the Nuvvuagittuq supracrustal belt, Canada. The study thus effectively combines real-world petrology with isotope geochemistry and modelling. I had some small queries about the preferred open vs. closed petrological modelling results and the interpreted geothermal gradient for TTG magma genesis, but my confusion may simply be due to the need to keep the word count down in the main text for this journal. A short expansion on these points would help me and presumably other readers to have a clear idea of the tectonic model put forwards, although Figure 4 is very effective in summarizing this. In short, I think that this is an excellent study and the results have a potentially high impact on our understanding of early Earth geodynamics, so it is well suited for publication in *Nature Communications*. Please do not mistake my brief review comments below for a lack of interest in the subject or a lack of attention while reading it – the work is well structured and edited, the data seem reliable, and the interpretations are defensible, so I had relatively little to say.

We thank the reviewer for his appreciation of our manuscript and his helpful comments that greatly aided our revisions.

Title: I rarely comment on titles, but I note that you explicitly mention evidence for plate tectonics, whilst being more circumspect (rightly so) in your conclusions (lines 190-193). Should the title be amended accordingly?

We agree with the reviewer and have changed the title to “Calcium isotope evidence for early Archaean carbonates and subduction of oceanic crust”.

Line 42: I would argue that this needs rewording. First, I assume you mean a combined TTG suite instead of the individual components, which of course may form individually by a variety of processes. Secondly, what is the “modern Earth”? Post-initiation of plate tectonics? The authors know this – I am quite sure – but TTGs likely formed in the early stages of a mobile lid regime when subduction was warm and slabs dipped at shallow angles (see geodynamic regime diagram below from Palin et al., 2020; *Earth-Science Reviews*). So, is the “modern Earth” defined by cold, steep, and deep subduction only? If so, when did *that* start?

We thank the reviewer for this comment and have now revised this sentence to clarify for the readers that (i) we mean TTG *suites* and (ii) that these no longer form “in the present day” (instead of “on modern Earth” for which it is hard to define a starting point) [line 42].

Line 102: Here (i.e. in Fig 2B), it seems quite fanciful to say that the data fit best *between* 500 °C/GPa and 750 °C/GPa, when in fact they look like they fit the 500 °C/GPa curve very well by themselves. More data points seem to lie on what would be the cold-temperature side of it than the high-temperature side. And it’s also worth pointing out that the dotted line representing open system (fractionation) of garnet during metamorphism of the DAT

appears to veer away from the measured TTG data considerably above ~1.5 GPa (so, 750 °C at this gradient), but the closed-system (equilibrium) scenario fits the TTG data quite well across the entire pressure and temperature range. Yet, you then show on Fig. 2 that TTG magma genesis occurred at temperatures above 700 °C, so is this the scenario you're championing? Closed-system melting? When does the final melt leave the system to form the plutons in the overlying crust? Is there a physical reason that it would stay in place after it exceeds the well-known melt connectivity threshold of ~7 vol. %? I'm purely thinking of the physical implications of your modelling results, here. Finally, how does this trace element analysis tie in to later plagioclase fractionation within the crustal magma chamber (lines 119-122) – this process wouldn't change the already-inherited Dy/Yb ratio, I assume? I realize that words are at a premium for *Nature* journals, but I think interpretation of the physical processes of melt formation and loss can be expanded upon slightly.

We thank the reviewer for this insightful comment and agree that most of the data in Fig. 2b fall on the predicted trend for 500°C/GPa, but we chose to be more conservative with our estimates after considering our model results for other trace-element ratios (*e.g.* Sr/Y) and major elements (CaO, Na₂O/K₂O, and Mg#) shown in Extended Data Fig. 3.

Regarding closed vs. open-system melting, we agree with the reviewer that our data appear to agree better with closed-system melting models. One possibility is that our samples do in fact record closed-system melting behaviour, which could be explained by thermal or rheological impediments to melt-loss that are only overcome when significant over-pressure is imposed upon the system (Havlin et al., 2013). Recent work suggests that most melt stays in the deep crust due to accumulation in dilational zones (Sawyer, 2020). Another possibility is that melt collects in specific zones at the edges of pre-existing diapirs, which have been shown to act as regions of very high melt collection (Zhang et al., 2020), prior to diking and emplacement of the magmas as TTG plutons. A final possibility is that competing open-system processes (*e.g.* garnet fractionation + melt loss) led to compositions similar to those predicted by closed-system models. Although there are too many unconstrained parameters for an additional combined garnet fractionation + melt loss model to be useful, we have now added a sentence to the main discussing the implications of our data agreeing best with closed-system models [lines 109-111].

The reviewer is correct that plagioclase fractionation in crustal plutons should not significantly affect the Dy/Yb ratios initially inherited from the melting process, we have now specified that this is the case [Lines 122-123].

Lines 113-114: I hate to bring this up, but Palin et al. (2016), who published these calculations and results a year earlier, should replace this reference (Johnson et al., 2017). The study referenced here performed almost identical work and – surprisingly – did not cite us (cf. the corrigendum written by Johnson et al. in response to a query on this issue made to the journal editor: www.nature.com/articles/nature22385). It is also worth noting that the bulk composition used by Johnson et al. (2017) is highly atypical of Archean depleted basalts (very low Mg#), as also acknowledged in their corrigendum, rendering their results of limited value for broad-scale interpretation of Archean processes.

- Palin, R.M., White, R.W. and Green, E.C.R., 2016. Partial melting of metabasic rocks and the generation of tonalitic–trondhjemitic–granodioritic (TTG) crust in the Archaean: Constraints from phase equilibrium modelling. *Precambrian Research*, 287, 73-90.

We thank the reviewer for this comment and have added a reference to Palin et al. 2016 at this point in the text [line 116].

Lines 114-116: All Archean TTGs or just the ones you are studying?

We thank the reviewer for this comment. Given that our samples have typical TTG compositions, span the entire Archean Eon, and represent a worldwide distribution, we argue that our data can be extended to Archean TTGs in general. We have now specified “Archean TTGs” in order to clarify this position [line 117].

Line 153: This should be “marly”, not “marley”, presumably?

We thank the reviewer for catching this mistake and have corrected the spelling of marly [line 159].

Lines 167-169: This seems a bit speculative, although the results of your analysis for the Nuvvuagittuq samples depends on it being true. Is there support for having the same average fractionation between sediment and seawater at this early point in Earth history?

We thank the reviewer for this comment and point him towards the last paragraph of the Supplementary Information (part 5b) where this topic is discussed. In short, calcite-seawater fractionation is hard to determine for this period in Earth history and theoretically depends (mainly) on the precipitation rates and the polymorph or carbonate that is being precipitated. Although biological carbonate production tends to increase precipitation rates, we note that experimental abiotic carbonate precipitation can exceed the ‘average’ carbonate-seawater fractionation (-1.25 per mil) in the absence of biology (Gussone et al., 2005; Tang et al., 2008). We have now added a clear indication in the main text that this is discussed in the supplementary information [line 175].

Lines 169-171: I didn’t catch whether this was weight % or volume %, but either way, you’re suggesting that up to half of the TTG magma analysed is assimilated shale plus carbonate sediment? That is a lot of sediment to transport down a subduction channel when most of the seafloor material is scraped away and forms accretionary prisms above the trench. Are there data to support this? Isn’t it also plausible that the sediment would dewater (or melt?) and transform into a metasediment at some point during burial *before* the subducted mafic crust melts?

We thank the reviewer for this comment and have now specified that our calculations give results in wt% [lines 178-179]. Although 30-50 wt% sediment incorporation can be considered high, up to 50% sedimentary materials have been previously suggested to explain Hf and O isotope signatures in granitic magmas (Kemp et al., 2007), and earlier iterations of our phase-equilibrium models suggest that lesser amounts of sediment incorporation (*e.g.* ~20 wt%) cannot produce melts with A/CNK high enough to explain the Nuvvuagittuq samples (A/CNK > 1.3).

One way to overcome the difficulties of incorporating metasediments into granitic melts is through the incorporation of previously melted sedimentary materials (Yakymchuk et al.,

2015). Another possibility is that low-T altered oceanic crust (e.g. hyaloclastites, containing both clays and carbonates, as opposed to marly sediments) is easier to subduct than unconsolidated sediments, but we believe that the specific mechanics of sediment incorporation during subduction/melting are beyond the scope of this study. We have now added several sentences to the supplementary information, however, discussing the possible scenarios that could have occurred during subduction of sediments [lines 379-388].

Line1 191-192: This is an important caveat to note (one instance of subduction \neq global plate tectonics), but how do your data show that “*subduction occurred continuously throughout the Archaean*”? Sorry f I missed something here.

We thank the reviewer for this comment and have now clarified this sentence to read “subduction events occurred repeatedly throughout the Archaean” [lines 204-205].

Lines 495-513: There are swaths of the Methods section that appear to be further discussion (i.e. not so much outlining the work that was done). This is a prime example and gives the worrying impression that some critical information or justification for the calculations performed is kept separate from the main text. I’d think about going through this Methods section to eliminate arguments for why you did what you did, and just stick to what you did (and how you did it)!

We agree with the reviewer and, given the short format of Nature group publications, have moved excessive discussion out of the Methods and into the Supplementary Information [now high-lighted in blue in the revised SI Text].

Supplementary Information

Lines 79-82: Can the authors explain why 5 mol % was used as a trigger point for performing fractionation? It seems arbitrary. Aren’t components sequestered into garnet continuously during growth, and doesn’t garnet grow discontinuously during metamorphism? Would the results differ if a step of 1 mol % or 10 mol % was used?

We thank the reviewer for this comment. The use of 5 mol% garnet as the size of the “batches” in our model is based on the phase-equilibrium modelling of (Kendrick and Yakymchuk, 2020). Given the high temperatures used in the modelling, continuous sequestration may not be realistic, particularly for major elements in garnet. The use of 1 mol% garnet increments would increase the magnitude of the effects as it yields results closer to those of a pure Rayleigh-fractionation model, while 10 mol% garnet would yield results closer to those of the equilibrium melting model. A statement to this effect has now been added to the methods section [lines 429-432].

Lines 127-243: This is another example of where there is extended discussion (i.e. line 134: “*In the following section, we provide a short review of modern geophysical observations for oceanic plateaus and adakitic magmatism*”). Although interesting, it seems like overkill when past studies can be cited to support the choice of geotherm that characterizes

different environments. If I put this another way, there are around 1600 words in this section alone (section 4 of the S.I.); nearly two thirds of the manuscript itself.

We thank the reviewer for this observation and have now shortened this section by several paragraphs (~200 words). We argue, however, that the thoroughness of this section is important because the information it contains guides our interpretations and has not yet been presented together by earlier works.

Lines 194-196: Hornblende is a candidate, though, and many studies have suggested that TTGs form due to melting of amphibolite or garnet amphibolite. Thus, you don't need trans-crustal hydrothermal systems if you're breaking down a hornblende-bearing rock and releasing structurally bound water.

We thank the reviewer for this comment and have removed the sentence in question from the shortened version of this section in the supplementary information, in accordance with his previous comment.

Extended Data Fig. 2: Are the lower panels showing the Ca weight % content of the minerals in question or the weight % of minerals that just contain Ca?

These panels show the contribution from each mineral to the total Ca budget of the rock, and thus consider both mineral abundances and Ca concentrations within each mineral. We have now clarified this in the figure caption [lines 27-28].

Reviewer #3 (Remarks to the Author):

The authors measured Ca isotopes in Archean granitoids and used the data to obtain some information about their origin and about the geodynamic regime that operated at this time. These ideas are novel and will they be of interest to many others in the community and the wider field. The new data seem to be of high quality and well worth publishing. However, I find many aspects of the interpretation unconvincing.

We thank the reviewer for his appreciation of our data and for his comments that have greatly improved the revised version of our manuscript.

1. In the introduction the authors equate subduction and plate tectonics; even though I think it is difficult to have one without the other, many others argue that demonstrating subduction does not imply the existence of rigid plates which, for them, are an essential component of plate tectonics.

We thank the reviewer for this comment and, also following the advice from reviewer #2, have now clarified that we are mainly discussing 'subduction' by replacing several key phrases throughout the text [line 23, line 31, line 46] that previously referred only to plate tectonics. We have also added a statement explaining that a global network of interconnected plate boundaries, in addition to subduction, is required by some definitions of 'plate tectonics' [line 204] and re-worded the title to reflect this distinction.

2. The correlations between $\delta^{44}\text{Ca}$ and the trace-element ratios that are used by the authors to infer PT trajectories are really rather poor. Three samples fall well above the line in Fig 2b; this feature is attributed to plagioclase fractionation. Five others fall below the line, which might imply low temperatures. For the Nuvvuagittuq samples, the low values are attributed to the presence of carbonate; the low- $\delta^{44}\text{Ca}$ of the others are not discussed. The position of the remaining 9 samples with intermediate $\delta^{44}\text{Ca}$ is shown to correspond to the modelled 500-750°C/GPa trajectory which is compared to that of a hot modern subduction zone. This is then taken as an argument that subduction operated at the time these granitoids were formed. This conclusion is not unreasonable (in my opinion, though others might disagree) but the line of argument seems rather tenuous.

We thank the reviewer for these helpful comments. Given that we have excluded possible kinetic and metamorphic effects as the source of the $\delta^{44}\text{Ca}$ variability, the only way to explain samples with positive $\delta^{44}\text{Ca}$ relative to BSE is through plagioclase fractionation, as this is the only known mineral to have a lower $\delta^{44}\text{Ca}$ than silicate melt. Regarding the 5 other samples that fall below our predicted trend for 500°C/GPa:

(i) Two of them (3-85 and 65-10) have strong positive Eu anomalies that likely represent plagioclase accumulation (Extended Data Fig. 3f), which would have decreased their $\delta^{44}\text{Ca}$ by up to $\sim 0.15\%$ [SI Text Section 5 (previously part of the Methods), Extended Data Fig. 5]. Thus, their compositions, prior to fractional crystallization, would agree with our model results for 500°C/GPa. We have now labelled these two points in Fig. 2b and specified in the figure caption that the associated discussion for these samples can be found in SI Text Section 5 [lines 132-133].

(ii) One of these samples (AG09-09) deviates only slightly from our models and agrees with some of our other model results in different parameter space (*e.g.*, $\delta^{44}\text{Ca}$ vs. Mg# and $\delta^{44}\text{Ca}$ vs. Eu/Eu* in Extended Data Figure 3). Although the slightly low $\delta^{44}\text{Ca}$ could potentially be explained through small amounts of carbonate incorporation, it may instead indicate formation along geotherms slightly lower than 500°C/Gpa [now stated explicitly, lines 139-140]; therefore, we do not believe that the evidence is strong enough to invoke sediment incorporation as an explanation for this sample.

(iii) The remaining two samples are the Nuvvuagittuq samples (INO5003 and INO5012) with compositions significantly different from all the other samples and from our model results. After careful consideration of the possible explanations (Supplementary information Section 6), these are most consistent with incorporation of clay-bearing/Ca-rich sediments (*e.g.* marlstone or hyaloclastite). We have now added a sentence to the main text specifying the reasons why these two samples are different than the other three samples that lie below our 500°C/GPa models [lines 137-140].

3. To explain the low $\delta^{44}\text{Ca}$ of the Nuvvuagittuq samples, the authors propose that the source contained carbonate sediments. For this to fit, they follow a rather tortuous path. Modern carbonates have only slightly negative $\delta^{44}\text{Ca}$ like those of the majority of the TTGs so the presence of this material can't explain the very low values of the Nuvvuagittuq

samples. The authors therefore propose that “the high internal heat-budget of early Earth ... promoted hydrothermally buffered Eoarchean seawater” to near BSE values, which would have led to strongly negative values in Eoarchean carbonate. This is not unreasonable, but it raises the question why this process did not operate during the formation of the other TTG, which have only slightly younger ages. Why was carbonate incorporated into the source of the Nuvvuagittuq samples but not that of the slightly younger samples, which formed at a time when the mantle was just as hot?

We thank the reviewer for this comment. This question is addressed in Section 7 of the Supplementary Information, where results from stable Si and Ca isotope analyses are compared. The short answer is that the difference is likely based on their location relative to settings where carbonates can accumulate (which depend on depth, latitude, and a variety of other factors) or specific conditions unique to subduction of Nuvvuagittuq protoliths. Although previous work suggests that chert was ubiquitously incorporated into TTGs (André et al., 2019; Deng et al., 2019), Ca isotopes suggest that carbonates must have either (i) been sparser, (ii) accumulated in settings less amenable to subduction, (iii) had $\delta^{44}\text{Ca}$ compositions similar to TTGs, and/or (iv) been more difficult to incorporate into TTG melts during subduction. We have now added a sentence to the main text that discusses these possibilities [lines 184-188].

4. The Nuvvuagittuq samples have elevated $A/CNK = \text{molar Al}_2\text{O}_3/(\text{CaO}+\text{Na}_2\text{O}+\text{K}_2\text{O})$ which is attributed to the presence of clay-rich sediment. However, a high A/CNK is difficult to reconcile with the presence of Ca-rich carbonate, which would decrease A/CNK . The authors claim to have found a recipe that works but this all seems rather contrived.

We thank the reviewer for this comment but disagree that high A/CNK is hard to reconcile with the presence of Ca-rich carbonates. There are several types of rock (*e.g.* marlstones, low-T altered oceanic crust) where clays and carbonates are found together. We have already accounted for the decrease in A/CNK when incorporating carbonates into shales in Fig. 3a, which is a three end-member mixing model including both types of sediments. These different types of sediments can also form separately and then be mixed together during subduction, as they are often found together in accretionary wedges/prisms above subduction zones (as discussed in Section 7 of the Supplementary Information and alluded to in the main text [lines 182-184]).

5. A role for carbonate in the Eoarchean granitoids is itself problematic, given that carbonate sediments are virtually absent from all Archean sequences, most probably due to the absence of shell-forming organisms and very different seawater compositions at that time.

We thank the reviewer for this comment. Due to the strong evidence for liquid water at Earth’s surface prior to 4.0 Ga [*e.g.* (Mojzsis et al., 2001; Wilde et al., 2001)], carbonate precipitation throughout the Hadean is thought to be necessary to maintain clement surface conditions (Hart, 1978). Without carbonate precipitation, the build-up of volcanic CO_2 in the atmosphere would have led to a runaway greenhouse effect on Earth. As the reviewer said,

however, massive carbonate units from this time have not been preserved in the rock record [although there is evidence for abiotic carbonate precipitation directly on the ocean floor in the form of aragonite fans (Grotzinger and James, 2000)].

The lack of carbonates in the rock record thus requires either (i) a re-evaluation of the evidence for liquid water and for the greenhouse gases that compensated for the faint young sun (Charnay et al., 2020; Kasting, 2010), or (ii) that carbonates were not preserved. We suggest that the latter is likely to be the case, which is one of the reasons that our findings are significant. We have now added a statement to this effect in the main text [lines 197-198].

An alternative explanation that might be explored is that terrestrial sediments are not involved; rather, the carbonate could have been a secondary phase in altered oceanic crust. Volcanic rocks in Archean greenstone belts commonly contain secondary carbonate. A distinctive feature of the Nuvvuagittuq rocks is that they formed at a time when very little continent was around and those continents that were around were largely flooded. Terrestrial sediment would have been virtually absent. The distinctive geochemistry of these rocks may reflect the presence in their source of carbonatized oceanic crust and deep-sea clay.

We thank the reviewer for this comment as we were not explicit in distinguishing the origin (e.g., oceanic vs. continental) of the sediments recorded by the Nuvvuagittuq samples. We agree carbonates and deep-sea clay in low-T altered oceanic crust could also explain our Nuvvuagittuq samples, which is why we suggested that our models would generally give the same results if we used hyaloclastites [e.g. (Martinez and Turi, 1978)] as the subducting component in the last two sentences of the methods section (originally lines 545-549, now moved to Section 6 of the supplement in line with comments from Reviewer #2). The reason we chose to use shales/carbonates as our sedimentary end-members is because there are much greater amounts of available data (e.g. Supplementary Information Table 5) that allow us to better constrain global averages. Nevertheless, these data agree well with sparser data for hyaloclastites (Martinez and Turi, 1978) and our interpretation in either case (that surface materials were recycled into the Nuvvuagittuq samples) remains the same. We have now specified that both oceanic (hyaloclastite) and continental (shale) sediments can account for our observations in the main text [line 179].

Comments from annotated PDF file:

Line 46: It is important to distinguish between subduction and plate tectonics. Many authors accept that some sort of subduction - shallow, discontinuous - operated in the early Archean but do not accept the presence of rigid plates which they consider to be the essential component of plate tectonics.

We thank the reviewer for this comment and, in line with comments from Reviewer #2, have changed the title of our manuscript as well as the associated text where there could be confusion between plate tectonics and subduction [line 23, line 31, line 46, line 204].

Line 50: Why? If we were to accept the melting-at-the-base-of-oceanic-crust model for granite formation (a silly idea, I admit), then surface-derived materials could have been buried down to the depths of magma formation. Cf. your line 57

We thank the reviewer for this comment but believe that there may be some confusion. We are saying that, although problematic for several geophysical reasons *e.g.* (Roman and Arndt, 2020), existing geochemical data for sediment incorporation into TTGs cannot differentiate between these two settings. However, our geothermal gradient constraints can be used for this purpose and serve as evidence against the previously proposed scenario of “top-down burial”, which is one of the reasons that our results are important.

Line 54: Really??? The trace element effect of garnet vs hbl are very different

We agree with the reviewer that the trace element effects of garnet and hornblende are not identical. Prior studies based dominantly on REE analyses (*e.g.*, Huang et al., 2020; Jagoutz et al., 2011; Kendrick and Yakymchuk, 2020; Liou and Guo, 2019; Smithies et al., 2019), however, cannot agree on a dominantly garnet- vs. hornblende-related signal, and thus this remains an issue for a number of authors.

Line 78: Except for the Acasta granitoid? IN fact I don't understand the whole phrase. For these >3 Ga granitoids, the near zero epsilon's Nd and Hf suggests low if any contamination. Why then do you mention assimilation of older crust?

We thank the reviewer for this comment but are left slightly confused. What we say in the manuscript (copied below) agrees with what he is saying: assimilation/crustal contamination was not a big factor in our samples. The Acasta tholeiitic granitoid (which is not a TTG) has slightly more negative ϵ_{Hf} values, but ϵ_{Nd} values within mantle range, also suggesting little to no crustal assimilation in this sample.

“Initial neodymium and hafnium isotopic compositions have limited ranges consistent with juvenile (mantle) sources and do not correlate with $\delta^{44}\text{Ca}$ (Figs. 1b, 1c), suggesting that assimilation of older crust did not influence isotopic signatures”

Line 79: These correlations are not at all convincing!

We thank the reviewer for this comment. Several processes can affect each of these trace-element proxies and can thus generate scatter in the data. Given that this is the case, we argue that having similar correlations (for many of our samples) between $\delta^{44}\text{Ca}$ and three different trace-element ratios that increase with pressure (Dy/Yb, Sr/Y, and La/Lu) is an important observation.

Line 149: Yes, but not carbonates!

We thank the reviewer for this short comment. He is correct that high A/CNK is not found in pure carbonates. That is why we use a three end-member mixing model that uses both shales (source of high A/CNK) and carbonates (source of high [Ca] and low $\delta^{44}\text{Ca}$) to explain our data. As suggested in one of the reviewer's earlier comments, we have now reiterated that oceanic sediments (*e.g.*, hyaloclastite) are also a viable candidate for explaining our observations [line 172].

REFERENCES

- André, L., Abraham, K., Hofmann, A., Monin, L., Kleinhanns, I.C., Foley, S., 2019. Early continental crust generated by reworking of basalts variably silicified by seawater. *Nat. Geosci.* 12, 769–773. <https://doi.org/10.1038/s41561-019-0408-5>
- Charnay, B., Wolf, E.T., Marty, B., Forget, F., 2020. Is the Faint Young Sun Problem for Earth Solved? *Space Sci. Rev.* 216, 90. <https://doi.org/10.1007/s11214-020-00711-9>
- Deng, Z., Chaussidon, M., Guitreau, M., Puchtel, I.S., Dauphas, N., Moynier, F., 2019. An oceanic subduction origin for Archaean granitoids revealed by silicon isotopes. *Nat. Geosci.* 12, 774–778. <https://doi.org/10.1038/s41561-019-0407-6>
- Grotzinger, J.P., James, N.P., 2000. Precambrian carbonates: evolution and understanding, in: *Carbonate Sedimentation and Diagenesis in the Evolving Precambrian World*. SEPM (Society for Sedimentary Geology), pp. 3–20. <https://doi.org/10.2110/pec.00.67.0003>
- Gussone, N., Böhm, F., Eisenhauer, A., Dietzel, M., Heuser, A., Teichert, B.M.A., Reitner, J., Wörheide, G., Dullo, W.C., 2005. Calcium isotope fractionation in calcite and aragonite. *Geochim. Cosmochim. Acta* 69, 4485–4494. <https://doi.org/10.1016/j.gca.2005.06.003>
- Hart, M.H., 1978. The evolution of the atmosphere of the earth. *Icarus* 33, 23–39. [https://doi.org/10.1016/0019-1035\(78\)90021-0](https://doi.org/10.1016/0019-1035(78)90021-0)
- Havlin, C., Parmentier, E.M., Hirth, G., 2013. Dike propagation driven by melt accumulation at the lithosphere–asthenosphere boundary. *Earth Planet. Sci. Lett.* 376, 20–28. <https://doi.org/10.1016/j.epsl.2013.06.010>
- Huang, G., Palin, R., Wang, D., Guo, J., 2020. Open-system fractional melting of Archean basalts: implications for tonalite–trondjemite–granodiorite (TTG) magma genesis. *Contrib. to Mineral. Petrol.* 175, 102. <https://doi.org/10.1007/s00410-020-01742-9>
- Jagoutz, O., Müntener, O., Schmidt, M.W., Burg, J.-P., 2011. The roles of flux- and decompression melting and their respective fractionation lines for continental crust formation: Evidence from the Kohistan arc. *Earth Planet. Sci. Lett.* 303, 25–36. <https://doi.org/10.1016/j.epsl.2010.12.017>
- Kasting, J.F., 2010. Faint young Sun redux. *Nature* 464, 687–689. <https://doi.org/10.1038/464687a>
- Kemp, A.I.S., Hawkesworth, C.J., Foster, G.L., Paterson, B.A., Woodhead, J.D., Hergt, J.M., Gray, C.M., Whitehouse, M.J., 2007. Magmatic and Crustal Differentiation History of Granitic Rocks from Hf-O Isotopes in Zircon. *Science* (80-.). 315, 980–983. <https://doi.org/10.1126/science.1136154>
- Kendrick, J., Yakymchuk, C., 2020. Garnet fractionation, progressive melt loss and bulk

- composition variations in anatectic metabasites: Complications for interpreting the geodynamic significance of TTGs. *Geosci. Front.* 11, 745–763.
<https://doi.org/10.1016/j.gsf.2019.12.001>
- Liou, P., Guo, J., 2019. Generation of Archaean TTG Gneisses Through Amphibole-Dominated Fractionation. *J. Geophys. Res. Solid Earth* 124, 3605–3619.
<https://doi.org/10.1029/2018JB017024>
- Martinez, M.P., Turi, B., 1978. The isotopic composition of oxygen and carbon in the hyaloclastites from the Mt. Iblei volcanic area, Eastern Sicily: A preliminary study. *Bull. Volcanol.* 41, 168–174. <https://doi.org/10.1007/BF02597220>
- Mojzsis, S.J., Harrison, T.M., Pidgeon, R.T., 2001. Oxygen-isotope evidence from ancient zircons for liquid water at the Earth's surface 4,300 Myr ago. *Nature* 409, 178–181.
https://doi.org/http://www.nature.com/nature/journal/v409/n6817/supinfo/409178A0_S1.html
- Roman, A., Arndt, N., 2020. Differentiated Archean oceanic crust: Its thermal structure, mechanical stability and a test of the sagduction hypothesis. *Geochim. Cosmochim. Acta* 278, 65–77. <https://doi.org/10.1016/j.gca.2019.07.009>
- Sawyer, E.W., 2020. Petrogenesis of Secondary Diatexites and the Melt Budget for Crustal Reworking. *J. Petrol.* 61, 1–23. <https://doi.org/10.1093/petrology/egaa039>
- Smithies, R.H., Lu, Y., Johnson, T.E., Kirkland, C.L., Cassidy, K.F.K.F., Champion, D.C.D.C., Mole, D.R.D.R., Zibra, I., Gessner, K., Sapkota, J., De Paoli, M.C.M.C., Poujol, M., 2019. No evidence for high-pressure melting of Earth's crust in the Archean. *Nat. Commun.* 10. <https://doi.org/10.1038/s41467-019-13547-x>
- Tang, J., Köhler, S.J., Dietzel, M., 2008. Sr²⁺/Ca²⁺ and ⁴⁴Ca/⁴⁰Ca fractionation during inorganic calcite formation: I. Sr incorporation. *Geochim. Cosmochim. Acta* 72, 3718–3732. <https://doi.org/10.1016/j.gca.2008.05.031>
- Wilde, S.A., Valley, J.W., Peck, W.H., Graham, C.M., 2001. Evidence from detrital zircons for the existence of continental crust and oceans on the Earth 4.4 Gyr ago. *Nature* 409, 175–178. <https://doi.org/10.1038/35051550>
- Yakymchuk, C., Brown, C.R., Brown, M., Siddoway, C.S., Fanning, C.M., Korhonen, F.J., 2015. Paleozoic evolution of western Marie Byrd Land, Antarctica. *Geol. Soc. Am. Bull.* 127, 1464–1484. <https://doi.org/10.1130/B31136.1>
- Zhang, N., Behn, M.D., Parmentier, E.M., Kincaid, C., 2020. Melt Segregation and Depletion During Ascent of Buoyant Diapirs in Subduction Zones. *J. Geophys. Res. Solid Earth* 125. <https://doi.org/10.1029/2019JB018203>

REVIEWERS' COMMENTS

Reviewer #2 (Remarks to the Author):

Review of: "Calcium isotope evidence for early Archean carbonates and subduction of oceanic crust"

The authors have done a good job addressing my comments and those of Reviewer 3. As I noted originally, I was supportive of publication and my comments were minor. Although the interpretations might not be readily accepted by everyone in the highly opinionated Archean community, the data, modelling, and tectonic interpretations presented here are defensible and worthy of entering the literature to initiate discussion. I look forwards to seeing the work in print.

Richard Palin
Associate Professor of Petrology, University of Oxford

Reviewer #3 (Remarks to the Author):

The authors have responded positively and constructively to the suggestions made by me and the other reviewer and I think the manuscript is now ready for publication. Nonetheless, I still find it difficult to accept the argument that sedimentary carbonate was incorporated in the Nuvvuagittuq samples and only into these samples and not the others with similar ages. Their hypothesis might be right, but I suggest that a more likely explanation is that the carbonate was within altered basaltic oceanic crust. The authors now mention that carbonated hyaloclastite might be involved and this is fair enough, but there is no real reason just to focus on hyaloclastite. Many basaltic flows in Archean greenstone belts are pervasively carbonated, the massive interiors of flows as well as the marginal zones, and although some of this alteration took place during metamorphism/metasomatism that followed accretion, and some of this alteration does seem to have taken place on the ocean floor. When one considers the question of CO₂ drawdown, a very large sink is required, one larger than just a thin layer of oceanic sediment. I think the authors have come up with good evidence of early subduction – namely that oceanic crust, altered at the surface, was incorporated into the source of granitoids – but the involvement of carbonate sediment is not necessary nor proven.

In the Nuvvuagittuq samples, the high A/CNK could well result from incorporation of 30-50% shale, as the authors show in their Figure 3A. I wonder, however, whether this incorporation happened not into the mantle source of the parental magmas, but during the within-crust melting that produced the granitoid magma. This would lead to a model in which the parental magma was formed through melting of a mantle source that contained carbonated basaltic crust; then this melt was contaminated with detrital sediment within the nascent continental crust.

This still does not answer the question of why only the Nuvvuagittuq samples show this signature. Perhaps the low $\delta^{44}\text{Ca}$ of the parental magmas of the other samples was camouflaged by the assimilation of other crustal material with more subdued $\delta^{44}\text{Ca}$. In any case I strongly recommend that the authors frankly discuss this issue.

Nicholas Arndt
Grenoble, January 2021

2nd Review Responses

Reviewer #2 (Remarks to the Author):

Review of: "Calcium isotope evidence for early Archaean carbonates and subduction of oceanic crust"

The authors have done a good job addressing my comments and those of Reviewer 3. As I noted originally, I was supportive of publication and my comments were minor. Although the interpretations might not be readily accepted by everyone in the highly opinionated Archean community, the data, modelling, and tectonic interpretations presented here are defensible and worthy of entering the literature to initiate discussion. I look forwards to seeing the work in print.

Richard Palin

Associate Professor of Petrology, University of Oxford

We thank the reviewer for his positive reviews and helpful comments that greatly improved our manuscript.

Reviewer #3 (Remarks to the Author):

The authors have responded positively and constructively to the suggestions made by me and the other reviewer and I think the manuscript is now ready for publication.

We thank the reviewer for his positive review and useful suggestions that greatly improved our manuscript.

Comment 1: Nonetheless, I still find it difficult to accept the argument that sedimentary carbonate was incorporated in the Nuvvuagittuq samples and only into these samples and not the others with similar ages. Their hypothesis might be right, but I suggest that a more likely explanation is that the carbonate was within altered basaltic oceanic crust. The authors now mention that carbonated hyaloclastite might be involved and this is fair enough, but there is no real reason just to focus on hyaloclastite. Many basaltic flows in Archean greenstone belts are pervasively carbonated, the massive interiors of flows as well as the marginal zones, and although some of this alteration took place during metamorphism/metasomatism that followed accretion, and some of this alteration does seem to have taken place on the ocean floor.

We thank the reviewer for this comment and have now specified in the main text that altered oceanic crust may be a possible (though unlikely) source for carbonates in the Nuvvuagittuq Supracrustal Belt (NSB) protoliths (lines 182-185), and added a more detailed discussion to the supplementary text (Section 6a, italic text copied below):

Carbonates within altered oceanic crust (AOC) could also be invoked to explain our data, but these tend to have more positive $\delta^{44}\text{Ca}$ than sedimentary carbonates (Blättler and Higgins,

2017), with Ca isotope compositions typically between BSE ($\delta^{44}\text{Ca} = 0\text{‰}$) and modern seawater ($\delta^{44}\text{Ca} = +0.9\text{‰}$), which could be due to (i) slower precipitation rates than for sedimentary carbonates and (ii) some of the Ca in AOC carbonates coming from basalts rather than purely from seawater. If the first scenario is correct, then it may be possible that BSE-like seawater with $\delta^{44}\text{Ca} = 0\text{‰}$ could have yielded AOC carbonates with $\delta^{44}\text{Ca}$ down to -0.9‰ , as required by our NSB samples, on the other hand, any Ca contributions from basalts would dilute this signal (pushing values back towards 0‰). Given the relatively smaller $\Delta^{44}\text{Ca}_{\text{SW-carbonate}}$ fractionations associated with AOC carbonates, an AOC origin for carbonates in NSB protoliths would require that (i) essentially all the Ca in the NSB samples came from carbonates and that (ii) these carbonates represented the lowest $\Delta^{44}\text{Ca}_{\text{SW-carbonate}}$ end-members observed for AOC (Blättler and Higgins, 2017). It is more likely, therefore, that smaller amounts of sedimentary carbonates with average $\delta^{44}\text{Ca}$ values lower than -0.9‰ (e.g. -1.25‰ , as suggested in the main text) were incorporated into the NSB protoliths.

Although subduction of sedimentary carbonates may be more difficult than AOC carbonates, Fe isotope and chemical data from NSB samples also suggest that sedimentary carbonates were present in NSB protoliths (Dauphas et al., 2007), while $\delta^{30}\text{Si}$ data in the same samples analyzed by this study (INO5003 & INO5012) suggest that incorporation of chemical sediments (chert) also occurred (Deng et al., 2019). Furthermore, $^{87}\text{Sr}/^{86}\text{Sr}$ measurements in zircon-hosted apatite from the same locality suggest that the NSB protoliths had high Rb/Sr ratios (Boehnke et al., 2018) potentially indicative of clay-rich sediments; and an associated terrane (the Ukaliq Supracrustal Belt, USB, $\sim 3\text{km}$ to the North) has relatively high $\delta^{18}\text{O}$ and $\delta^{30}\text{Si}$ values also suggestive of chemical sediments in protoliths (Chowdhury et al., 2020). Indeed, the large number of metasedimentary units associated with amphibolites in the NSB region (Cates and Mojzsis, 2009, 2007; Greer et al., 2020; Mloszewska et al., 2013, 2012), which are potential precursors to TTGs in the area (Adam et al., 2012), are likely to have been assimilated into TTG melts. Together, these observations lend weight to our hypothesis that both terrigenous sediments (shales) and chemical sediments (carbonates) were, at least locally, present in the Eoarchean oceans and were subsequently incorporated into NSB TTGs.

The question as to why the NSB samples require incorporation of carbonates, while other TTG samples from the early Archaean do not, is addressed in a subsequent response (Comment 4).

Comment 2: When one considers the question of CO₂ drawdown, a very large sink is required, one larger than just a thin layer of oceanic sediment. I think the authors have come up with good evidence of early subduction – namely that oceanic crust, altered at the surface, was incorporated into the source of granitoids – but the involvement of carbonate sediment is not necessary nor proven.

We thank the reviewer for this comment and agree that a large CO₂ sink is required to maintain clement surface conditions on the early Earth. We also agree that the light $\delta^{44}\text{Ca}$ signatures in NSB samples may potentially reflect a process-based (e.g. Ca diffusion into NSB TTGs) rather than source-based mechanism; however, we have carefully considered all the possibilities and argue that the most parsimonious explanation, taking into account all of our data (and previous work), is that carbonate and clay sediments were incorporated into NSB protoliths.

Comment 3: In the Nuvvuagittuq samples, the high A/CNK could well result from incorporation of 30-50% shale, as the authors show in their Figure 3A. I wonder, however, whether this incorporation happened not into the mantle source of the parental magmas, but during the within-crust melting that produced the granitoid magma. This would lead to a model in which the parental magma was formed through melting of a mantle source that contained carbonated basaltic crust; then this melt was contaminated with detrital sediment within the nascent continental crust.

We thank the reviewer for this comment and acknowledge that assimilation of shales with high A/CNK could have possibly occurred during emplacement of the TTG plutons. However, this is less likely than incorporation through subduction, because it would require very large heat fluxes (to assimilate 30-50 wt% sediments into the melts by AFC). Assimilation of this large quantity of sedimentary material is a self-limited process given the thermal requirements of assimilation (e.g. Glazner, 2007), especially in relatively low-temperature granitoid magmas. Given that there is also previous evidence for incorporation of sediments in NSB samples, we argue that both shale and sedimentary carbonates were subducted prior to formation of TTG plutons in the region. We have now added a statement to this effect in the supplementary text (Section 6b, lines 414-418).

Comment 4: This still does not answer the question of why only the Nuvvuagittuq samples show this signature. Perhaps the low $\delta^{44}\text{Ca}$ of the parental magmas of the other samples was camouflaged by the assimilation of other crustal material with more subdued $\delta^{44}\text{Ca}$. In any case I strongly recommend that the authors frankly discuss this issue.

We thank the reviewer for this comment and acknowledge that, with our current data, we cannot disprove that other TTG samples also incorporated carbonate sediments, and that their $\delta^{44}\text{Ca}$ signatures were then masked by other processes. One possibility (discussed in the main text) is that a majority of early Archaean carbonates had $\delta^{44}\text{Ca}$ values similar to TTGs [e.g., -0.3‰, which would represent slow (equilibrium) precipitation from BSE-like seawater]. Nonetheless, sediments are not *necessary* to explain the geochemical compositions of our other samples (as shown by our phase-equilibrium model results and now reiterated in the main text, lines 193-197). The only samples with an obvious sedimentary influence (high $\delta^{18}\text{O}$, high A/CNK, and anomalously low $\delta^{44}\text{Ca}$) in our current data set are those from the NSB, which is why we focus our discussion on these samples. Although we cannot currently conclude that carbonate incorporation into TTGs was a global process, future work may allow for the eventual identification of more cryptic sedimentary carbonate signatures in TTG samples. In any case, whether carbonate incorporation into TTG magmas is specific to NSB or global does not affect our major conclusion, even though it is an important question.

Nicholas Arndt

Grenoble, January 2021

References

Adam, J., Rushmer, T., O'Neil, J., Francis, D., 2012. Hadean greenstones from the Nuvvuagittuq fold belt and the origin of the Earth's early continental crust. *Geology* 40, 363-366.

<https://doi.org/10.1130/G32623.1>

- Blättler, C.L., Higgins, J.A., 2017. Testing Urey's carbonate–silicate cycle using the calcium isotopic composition of sedimentary carbonates. *Earth Planet. Sci. Lett.* 479, 241–251. <https://doi.org/10.1016/j.epsl.2017.09.033>
- Boehnke, P., Bell, E.A., Stephan, T., Trappitsch, R., Keller, C.B., Pardo, O.S., Davis, A.M., Harrison, T.M., Pellin, M.J., 2018. Potassic, high-silica Hadean crust. *Proc. Natl. Acad. Sci.* 115, 6353–6356. <https://doi.org/10.1073/pnas.1720880115>
- Cates, N.L., Mojzsis, S.J., 2009. Metamorphic zircon, trace elements and Neoproterozoic metamorphism in the ca. 3.75 Ga Nuvvuagittuq supracrustal belt, Québec (Canada). *Chem. Geol.* 261, 99–114. <https://doi.org/10.1016/j.chemgeo.2009.01.023>
- Cates, N.L., Mojzsis, S.J., 2007. Pre-3750 Ma supracrustal rocks from the Nuvvuagittuq supracrustal belt, northern Québec. *Earth Planet. Sci. Lett.* 255, 9–21. <https://doi.org/10.1016/j.epsl.2006.11.034>
- Chowdhury, W., Trail, D., Guitreau, M., Bell, E.A., Buettner, J., Mojzsis, S.J., 2020. Geochemical and textural investigations of the Eoarchean Ukaliq supracrustals, Northern Québec (Canada). *Lithos* 372–373, 105673. <https://doi.org/10.1016/j.lithos.2020.105673>
- Dauphas, N., Cates, N.L., Mojzsis, S.J., Busigny, V., 2007. Identification of chemical sedimentary protoliths using iron isotopes in the > 3750 Ma Nuvvuagittuq supracrustal belt, Canada. *Earth Planet. Sci. Lett.* 254, 358–376. <https://doi.org/10.1016/j.epsl.2006.11.042>
- Deng, Z., Chaussidon, M., Guitreau, M., Puchtel, I.S., Dauphas, N., Moynier, F., 2019. An oceanic subduction origin for Archaean granitoids revealed by silicon isotopes. *Nat. Geosci.* 12, 774–778. <https://doi.org/10.1038/s41561-019-0407-6>
- Glazner, A.F., 2007. Thermal limitations on incorporation of wall rock into magma. *Geology*, 35, 319–322. <https://doi.org/10.1130/G23134A.1>
- Greer, J., Caro, G., Cates, N.L., Tropper, P., Bleeker, W., Kelly, N.M., Mojzsis, S.J., 2020. Widespread poly-metamorphosed Archean granitoid gneisses and supracrustal enclaves of the southern Inukjuak Domain, Québec (Canada). *Lithos* 364–365, 105520. <https://doi.org/10.1016/j.lithos.2020.105520>
- Mloszewska, A.M., Mojzsis, S.J., Pecoits, E., Papineau, D., Dauphas, N., Konhauser, K.O., 2013. Chemical sedimentary protoliths in the >3.75Ga Nuvvuagittuq Supracrustal Belt (Québec, Canada). *Gondwana Res.* 23, 574–594. <https://doi.org/10.1016/j.gr.2012.11.005>
- Mloszewska, A.M., Pecoits, E., Cates, N.L., Mojzsis, S.J., O'Neil, J., Robbins, L.J., Konhauser, K.O., 2012. The composition of Earth's oldest iron formations: The Nuvvuagittuq Supracrustal Belt (Québec, Canada). *Earth Planet. Sci. Lett.* 317–318, 331–342. <https://doi.org/10.1016/j.epsl.2011.11.020>